# A Non-Sacrificial 3D Printing Process for Fabricating Integrated Micro/Mesoscale Molds

**DOI:** 10.3390/mi14071363

**Published:** 2023-06-30

**Authors:** Amirreza Ghaznavi, Jie Xu, Seth A. Hara

**Affiliations:** 1Department of Mechanical and Industrial Engineering, University of Illinois at Chicago, Chicago, IL 60607, USA; aghazn2@uic.edu (A.G.); jiexu@uic.edu (J.X.); 2Division of Engineering, Mayo Clinic, Rochester, MN 55905, USA

**Keywords:** microfluidics, mold fabrication, 3D printing, micro/mesoscale molds

## Abstract

Three-dimensional printing technology has been implemented in microfluidic mold fabrication due to its freedom of design, speed, and low-cost fabrication. To facilitate mold fabrication processes and avoid the complexities of the soft lithography technique, we offer a non-sacrificial approach to fabricate microscale features along with mesoscale features using Stereolithography (SLA) printers to assemble a modular microfluidic mold. This helps with addressing an existing limitation with fabricating complex and time-consuming micro/mesoscale devices. The process flow, optimization of print time and feature resolution, alignments of modular devices, and the advantages and limitations with the offered technique are discussed in this paper.

## 1. Introduction

Three-dimensional printing technology has recently emerged as a promising method for fabricating microfluidic and other lab-on-a-chip devices due to its ability to rapidly prototype unique three-dimensional structures [1,2]. Traditional fabrication processes such as lithography and micromachining can be labor intensive, time consuming, and expensive. With 3D printing, engineers can use computer-aided design (CAD) software to specify part geometry and design complex 3D features [3,4]. There are two general approaches implemented for fabricating microfluidic devices using 3D printing technology: directly 3D printing the device or indirectly using the 3D-printed molds for soft lithography processes [5,6,7,8]. Studies have shown the feasibility of taking either of the approaches to produce monolith or hybrid microfluidic devices using various materials and techniques in 3D printing [9,10,11]. Ruiz et al. [12] showed the integration of Fused Deposition Modeling (FDM) and Stereolithography (SLA) 3D printing to produce a finger-actuated pump, a microfluidic quick connect component, and a microfluidic reactor chip. In their study, the feasibility of integrating two different 3D printing techniques, SLA and FDM, to produce modular microfluidic devices was demonstrated, and the functionality of the devices was discussed. In another study, a FDM 3D printer was used to fabricate complicated microfluidic scaffolds with ABS, which is soluble in acetone. They could fabricate complicated channels in PDMS by casting PDMS on the 3D-printed scaffolds and dissolving the sacrificial scaffolds with acetone [13,14]. Chan et al. [15] printed the soft-lithography master to fabricate PDMS microfluidic devices. In their study, the post-treatment of 3D-printed masters was investigated to facilitate their functionality as masters for PDMS replica molding. Additionally, Digital Light Processing (DLP) technology has been utilized along with FDM and SLA 3D printing technologies as a fabricating tool in order to produce microfluidic molds and devices [1,16,17].

Three-dimensional printers have enabled engineers and scholars to print high-resolution features, as small as one micron, thanks to the advancement in technology in recent years [18,19,20]. The printers which are capable of printing high-resolution features are normally expensive due to the resolution offered, and the printing time is mostly long since printing with high resolution requires precise curing of the materials on top of previously printed layers [21,22]. On the other hand, 3D printers which mostly offer lower resolutions have shorter print times for mesoscale features due to the lower print precision required for the mesoscale parts, and the cost of these printers is lower than that of high-resolution printers since the hardware used has lower resolution specifications [23,24]. Therefore, there is an existing challenge with producing devices with microscale (MICS) and mesoscale (MESS) features as a monolith or hybrid device using 3D printing technology [25,26].

To overcome this challenge and connect micro (less than 100 μm) to mesoscale (more than 100 μm up to few millimeters) features in a microfluidic device, an integration of both high-resolution and conventional 3D printers to manufacture parts, which can be assembled to form composite microfluidic molds, is offered in this study. By sectioning the design file into two parts, MICS and MESS, parts can be printed separately, and the part with MICS features can be assembled with the MESS part to form a composite device. Thus, not only can the print time and cost of the 3D-printed microfluidic mold be optimized, but the micro- and meso-features would also be integrated into a single device. Through this method, the fabrication of complicated molds using 3D printing technology would be facilitated for researchers and engineers. Previously, molds with micro-features were fabricated using traditional processes such as micromachining using CNC drilling machines or through the multiple steps of lithography processes, including photolithography, etching (wet or dry), etc.; however, these traditional approaches usually take multiple steps and require expertise and clean-room facilities [18,27]. By implementing high-resolution and conventional 3D printers, a non-sacrificial 3D-printed mold fabrication process [28,29] is offered to help researchers with the quick fabrication of simple and complex molds for applications such as microfluidics. Three-dimensional-printed micromixers have been widely used in microfluidics applications including but not limited to chemical synthesis, drug delivery, and medical diagnosis [16,30,31]. To prove the feasibility of implementing our approach in fabricating microfluidic molds, we designed and fabricated a micromixer and used food dyes to demonstrate the profile of mixed liquids in the device. The integration of the 3D-printed MICS features with the MESS 3D-printed part is not only limited to the mixing applications; however, it is a solid proof of concept for the offered technique in mold fabrication.

## 2. Materials and Methods

For printing the MESS part for a microfluidic mold, Clear V4 resin (Formlabs, Somerville, MA, USA) was used in a Form 3B (Formlabs) printer. The MICS part of the mold was printed with a high-resolution SLA 3D printer from Kloé SA (Dilase 3D) using clear resin (DS3000, DWS, Thiene, Italy). For both printers, post-processing was performed by first immersing the parts in isopropyl alcohol (IPA) for 15 min and then globally UV-curing them for 15 min. To measure the dimensions of printed parts, a digital microscope (Keyence VHX-7000, Keyence Corporation, Illinois, IL, USA) was used, and the measurement was conducted before and after mating parts. Moreover, the pictures and the surface profiles of the devices printed using both 3D printers were obtained using the microscope. The fabrication of PDMS was conducted using SYLGARD 184 (Dow Corning, Midland, MI, USA) with a 10:1 ratio of polymeric base and curing agent. After mixing these two components, the mixture was poured on the mated 3D-printed molds and degassed for three hours to remove the air bubbles. Lastly, the device was put in the oven and heated to 70 degrees centigrade for two hours.

## 3. Results and Discussion

### 3.1. Process Flow

Understanding the process flow of this novel approach is important since it helps with the precise design and fabrication of microfluidic molds that leads to a facile and concise mating procedure, Figure 1. To demonstrate the proposed approach and process flow, a serpentine mixer mold with MICS features is presented. This mixer mold not only consists of channels with MICS features but also features, including channels, inlets, and an outlet, with MESS dimensions.

To start, a CAD file of the desired part is required for 3D printing. The mixer mold was designed with two channel widths of 150 and 500 μm (Appendix A) and inlet and outlet diameters of 3 mm. Considering the CAD model, the features with MICS features were delineated and separated from the MESS ones in the second step. The resolution of the MESS/MICS printers should be considered in this step since it informs the delineation process of MESS/MICS features. Thus, a set of calibration tests needs to be conducted to understand the resolution limitations of MESS/MICS printers. Calibration tests should be selected that best represent the features of interest for a particular model. For the mixer mold, calibration tests were performed to understand the limitations regarding accurately printing straight channels, and the corresponding data are presented in Section 3. Using these data, the 150 μm channels were delineated from 500 μm since the resolution of the MESS printer could adequately print the larger channels but not the smaller ones. Moving to the next step, micro-features should be sectioned from meso-features. An important consideration in this step is leaving a few millimeters to centimeters offset from the border of the micro-features. This helps with the manual mating process since delicate features would be safe and intact from any possible damage caused by the manual handling of the MICS part. For the mixer mold, the sectioned part for MICS features included the entire serpentine channels in a 5 by 5 mm area with a z height of 500 μm. This part would be printed by the MICS printer, and the rest of the mold would be printed using the MESS printer.

Once the CAD file is sectioned, the MESS part (in this work, the base in the mixer mold) is printed and post-processed, as detailed in the Materials and Methods. The next step is measuring the critical dimensions of the base and modifying the insert part, which has micro-features, accordingly. To measure the dimensions accurately, a high-resolution microscope was used to capture clear images at different heights along the *z*-axis. This helped with the measurement of critical layers for aligning the printed parts such as the bottom and top layers of the base, the top surface of MESS features, and other features where MESS/MICS parts interface. The steps mentioned in this paragraph were applied to the mixer shown in Figure 2.

The measurements taken from the MESS base should be applied in the design of the MICS part(s) (the insert in mixer mold). For instance, if a printed dimension of the MESS base was smaller than designed, the interfacing portion of the MICS insert was reduced to accommodate the MESS print error. In addition, the geometrical errors caused during the printing of MESS part, such as rounded corners or non-vertical sidewalls, can be applied into the design modification of the MICS part. This step helps facilitate the process of mating and assembling the sectioned parts of the mold since modifying the part dimensions results in removing sharp edges and improving surface properties at the intersecting areas of the MICS and MESS parts. Once design modifications were applied to the MICS part design, the part was printed with the MICS printer and subsequently post-processed based on the procedure in the Materials and Methods. Then, different dimensions of this part, such as length, width, and corner radius, were measured using the same microscope used for the base. This helped to check the quality of the print with regards to defects, accuracy of dimensions, and the orientation of features which need to be aligned with the base according to the design. To mate parts, tweezers were used to handle the insert and place it in the base. Once the alignment was confirmed with a microscope, the tip of the tweezers was used to apply uniform force and push the insert into the base for a friction fit. The mated parts were inspected to check the alignment of features after mating.

### 3.2. Comparison of 3D Printers’ Resolution

The resolution of a 3D printer can be a limiting factor in printing target geometries in microfluidics applications, especially using MICS 3D printing. In microfluidics, feature dimensions can significantly impact the behavior of fluids inside microchannels, for example, the mixing profile of fluids for mixing applications or the interaction of fluid components with each other and the device itself, such as the alignment of cells in microwells or the docking process in sorting applications. Therefore, identifying the limitations of 3D printers in printing different sizes and geometries of features is a step towards the design of CAD files (first step in Figure 1). A calibration process is required before moving on with the process flow discussed in Section 1. In this section, a simple calibration process which could help with identifying the orientation of parts on the build platform, channel surface profiles, and geometrical accuracy and limitations of micro/meso-SLA 3D printers, is presented. By this way, the design and fabrication of the mixer mold would be more accurate since the limitations of the 3D printer are known.

To understand the capabilities of the MESS (Formlabs 3B) and MICS (Kloe Dilase 3D) 3D printers, straight rectangular prisms were designed in different widths to quantify the resolution of printed features. For the MICS printer, straight rectangular prisms with widths of 50, 100, 200, and 300 μm, and for the MESS printer, channels with widths of 100, 200, 300, 400, 500, 600, 750, and 1000 μm were designed and printed separately. The length of the rectangular prisms was 10 mm for the MESS features and 3 mm for the MICS features. In Table 1, the measured dimensions, three samples and five measurements for each sample, for the printed features are shown. Based on the results obtained, the 150 µm serpentine channels of the mixer mold were delineated to be created by the MICS printer and the 500 μm channels were created by the MESS printer. It is possible to have smaller and more complicated channels printed with the MICS printer; however, this would require more process development to accurately print high-aspect-ratio channels and a longer degassing and curing process for PDMS.

According to the measurement data collected in the calibration process of straight channels (Table 1), the error percentage for almost all the designed dimensions is in an acceptable range (below 5%), except the channel with a 100 μm width printed with the MESS printer, delineating the resolution limitation of the MESS printer. Therefore, printing channels with a finer width (200 μm and below) should be printed with the higher-resolution printer (MICS). The straight channels printed with both printers are shown in Figure 3.

As part of the calibration procedure, the orientation of parts on the build plate of the MESS printer was investigated in this work. Different angles, including 0, 15, 30, 45, 70, and 90 degrees, were considered for orienting the mixer mold (Appendix A). Then, the channel widths and side profile widths were measured at different points based on their orientation angle. The collected data from the measurement are shown in Table 2, and the surface profile of the serpentine channels was obtained by using the integrated profilometry system of the digital microscope. The surface profilometry was only feasible to conduct for the MESS parts printed in Formlabs 3B due to the surface finish and white light reflection properties of the clear resin used for these parts. MICS parts could not provide any data for the surface profilometry since the surface finish was smoother and the resin used was clearer than in the MESS parts; thus, light reflection from the surface could not be detected by the integrated scanner of the microscope. Based on a visual inspection of the MICS parts (Appendix A), the surface finish was smoother than that of the MESS parts.

**Table 2 micromachines-14-01363-t002:** Measured dimensions of mixer channels printed using the MESS printer. Mean ± standard deviation. Error from the designed length is given in parentheses after the mean value. The designed channel width is 500 µm. Measured features are defined in Figure 4. Three samples were printed for each orientation, and five measurements were taken for each sample. NM = Not Measured, ND: Not Designed.

Feature	Designed Length (μm)	0 Degree	15 Degree	30 Degree	45 Degree	70 Degree	90 Degree
a	500	514.1 ± 5.23 (2.82%)	493.35 ± 10.38 (1.33%)	477.8 ± 7.62 (4.44%)	475.53 ± 5.65 (4.89%)	486.2 ± 18.91 (2.76%)	456.13 ± 4.5 (8.77%)
b	ND	82.33 ± 5.06	103.35 ± 11.2	116.93 ± 9.81	135.8 ± 8.8	141.5 ± 13.15	183.26 ± 0.57
c	ND	88.42 ± 8.2	114.7 ± 9.7	125.8 ± 6.77	138.53 ± 4.34	153.26 ± 6.19	172.4 ± 7.71
d	500	509.3 ± 12.75 (1.86%)	501.66 ± 8.9 (0.33%)	487.73 ± 1.22 (2.45%)	428.46 ± 40.99 (14.3%)	421.2 ± 20.8 (15.76%)	463.6 ± 35.02 (7.28%)
e	ND	85.46 ± 7.32	133.4 ± 2.9	188.13 ± 5.16	244.00 ± 8.9	363.8 ± 20.03	462.0 ± 9.23
f	ND	92.66 ± 7.85	85.40 ± 10.4	73.08 ± 8.92	125.95 ± 43.69	92.41 ± 28.74	NM
g	500	529.33 ± 11.01 (5.86%)	495.13 ± 43.24 (0.97%)	501.5 ± 7.62 (0.3%)	494.75 ± 19.08 (1.05%)	476.26 ± 12 (4.74%)	463.41 ± 9.38 (7.31%)
h	ND	97.5 ± 4.59	109.58 ± 20.17	104.53 ± 16.86	118.71 ± 9.85	172.13 ± 34.61	137.36 ± 14.94
i	ND	97.5 ± 4.24	125.75 ± 1.76	162.93 ± 5.81	181.41 ± 14.91	198.79 ± 85.61	NM

Table 2 presents the measurements (three samples for each angle and five measurements for each sample) for the channel width and side profile widths of serpentine channels printed using the MESS printer. This table helped with identifying the best angle for orienting the mixer mold on the build platform since the dimensions of the printed features are significantly important in the proposed technique of fabricating composite molds due to the translation of surface profiles and feature sizes to the final PDMS device. Since the least error is desirable for producing composite MICS/MESS features, the 0-degree orientation on the build platform was chosen to print composite parts in MESS printer (mold base). This is concluded based on the relative error percentages of the 500 μm channel width (critical dimensions a, d, and g) with 0 to 90 degrees, which are shown in Table 2 in parentheses right by the measured dimensions, and the side profile dimensions (b, c, e, f, h, i dimensions). By increasing the orientation angle from 0 to 90, the channel width decreases since the layers tend to bleed over while UV light exposes the target layer, and as a result, the side profile width increases. The critical dimensions (a, d, and g) of the mixer mold with a 15-degree orientation on the build platform showed less relative error percentages than the 0-degree orientation; however, by comparing the side profile dimensions of these two angles, it is preferable for the molding application of serpentine channels to have smaller side profile dimensions (with vertical walls), since the side profile of features directly translates to the generated PDMS device with angular walls (compromised features and dimensions). In Figure 5B, the 0-degree orientation shows the lowest average side profile dimension of channels compared to the rest of the orientations from 15 to 90 degrees.

### 3.3. Print Time

One of the main considerations in 3D printing, especially in MICS printing, is the print time. This factor significantly depends on the dimensions of the part and its orientation on the build platform since the process of SLA 3D printing is intrinsically dependent on the speed of the laser in the x or y directions and the height of each layer [30]. In addition, another factor to be considered in the total running time of a part is the degradation of materials used in the 3D printer, such as the build tank and resin, that occurs with use. Therefore, the printing time should be taken as a serious factor in the design and delineation of MICS/MESS features in our proposed approach.

As mentioned in the Introduction, printing features with MESS/MICS dimensions using a conventional SLA 3D printer will result in the poor resolution of the MICS features. On the other hand, using a high-resolution micro-SLA 3D printer will result in a prohibitively long print time due to the small size of the laser beam (mostly below 50 μm) and the height of each layer, ranging from 100 μm to hundreds of nanometers, based on the desired surface finish and the total time of print. Thus, there is a need to address the challenge of the print time of MICS/MESS parts. As it is shown in the process flow of the proposed technique in Figure 1, the most important step for optimizing the print time is where micro-/meso-features are delineated and sectioned from the CAD file. In this step, considering the height of features which have MICS dimensions plays a significant role in the print time since they can be printed with a thinner slicing height, resulting in a longer print time. A 50 μm slicing height was chosen for the mixer base and insert as this slicing height provided an acceptable resolution and print time in both printers. In addition to the slicing height, the total height of the sectioned MICS part (the insert) should be large enough to facilitate the handling of parts during assembly. If the insert part’s height is not tall enough to sit well in the base part, there would not be enough surface area to provide sufficient friction to keep the insert in place and it would come off from the base during the procedure of peeling off the PDMS from the assembled mold, and as a result, the mold could only be used a single time.

Since delineating the MICS from MESS features and considering enough spacing for sectioning are crucial for improving the running print time, the serpentine channels (MICS features with 400 μm height) in the mixer mold were sectioned with a 500 μm depth from the surface of the channels’ base so that the total height of the insert part would come to 900 μm. As we sectioned the MICS part based on the information provided, the estimated total print time of the mixer mold based on a 50 µm slicing height is shown in Table 3. It can be understood from the table that printing the entire mold with the MICS printer makes the printing impossible due to the degradation of the release layer of the tank. This release layer is made of PDMS, and exposure to UV light for a certain time (approximately 10 h) leads to its degradation and results in compromised print quality. On the other hand, printing the entire mold, including the insert and base parts, with the MESS printer would result in the poor resolution of serpentine channels (Figure 6a). Therefore, following the protocol discussed in the Process Flow section with regards to delineating the MESS/MICS parts and sectioning them appropriately, not only would the total print time of the composite mold significantly decrease compared to other approaches, but also the final resolution of the mold would be improved.

### 3.4. Alignment of Composite Parts

Once the best orientation of the mixer mold on the build platform was identified and the capabilities of the micro-/meso-printers in printing straight channels were understood, those results were used to print and mate composite parts. To evaluate the best alignment for the MICS and MESS parts, the critical dimensions of the base were measured for three samples with five measurements per sample, and four different levels of clearance, negative values (meaning smaller than the actual measured dimension) and positive values (bigger than the actual measured dimensions), were applied to the CAD design file for the insert part. The inserts were printed with these clearances and assembled with bases to evaluate the optimal clearance to ensure proper alignment between the MICS and MESS parts. Considering different clearances helped with understanding the best alignment of the composite parts since any defects from the misalignment could compromise the quality of the resulting PDMS device and damage features at the interface of the MICS and MESS parts. In Figure 7, the misalignments inspected in mated parts are defined, and the magnitude of misalignment for each feature is presented in Table 4.

According to the data presented in Table 4, a −10 μm clearance for the insert showed the minimum gap between the MESS and MICSS channels and minimum *Z*-axis misalignment, which was approximately equivalent to the height of one layer (50 μm). With regards to base *Z*-axis misalignment and *X*-axis misalignment, this clearance magnitude ranked second among all five clearance values considered for the alignment of composite parts in the mixer mold. As shown in Table 4, only two clearance values, −10 and −50 μm, showed a gap at the interface of the MESS and MICS parts (Figure 7c). Based on the results obtained from the misalignment of composite parts with different clearances, the best clearance, −10 μm clearance, which showed least misalignments, was considered to modify and improve the misalignment of the MICS and MESS parts. The *Z*-axis misalignments of the MICS part was the target parameter of this improvement since the magnitude of misalignment was considerably larger in both *Z*-axis misalignments than in the other measured features. This was likely due to microscope limitations with depth measurement when the base (MESS part) was measured for its critical dimensions. Thus, to improve the Z alignment, a layer from the last layer of serpentine channels for a set of molds and for another set one layer from the last layer of serpentine channels and one layer from the base of the insert part (MICS) were reduced. This modification was introduced to reduce the 50 and 100 µm z-height and resulted in an improvement in all misalignments measured, as shown in Figure 8. The insert part with two layers subtracted showed less misalignments in all measured features than the part with one layer subtracted. Therefore, this design and fabrication process was selected to fabricate the final PDMS mixer mold.

### 3.5. PDMS Molding and Characterization

After characterization of the 3D-printed assembly, the micro-/mesoscale mixer mold was rinsed with DI water and cleaned with a pressurized nitrogen blast to remove any dust particles on the surface. Then, the degassed and uncured PDMS was cast on the mold, and the degassing process was repeated in order to remove any air bubbles. After 1 h of degassing, the PDMS was cured in a preheated oven at a temperature of 80 centigrade for 3 h. To generate the final PDMS serpentine mixer, the PDMS was peeled off from the mold and adhered to a clean glass slide by utilizing plasma surface treatment. It was observed that the PDMS peeled from the 3D-printed mold with no defects or cracks (Figure 9).

The extracted PDMS was inspected to understand the translation of geometrical features, including the serpentine microchannels and associated side profile, into the PDMS (Figure 10 and Figure 11). According to the conducted measurements of three MICS parts and three PDMS samples (Table 5), the channels printed with the MICS 3D printer had a tapered side profile since the measurements of features on the top surface showed smaller magnitudes compared with the measured features on the bottom surface. This is clear from the inspection of PDMS devices after peeling off from the MICS part and their corresponding measurements, Figure 10D. However, the top surface of PDMS, where PDMS meets the glass slide, demonstrated a flat surface, which provided good adhesion of the PDMS device to the glass slide. Since the modified PDMS mold, the mold with two layers subtracted, was used for the final molding, the misalignment in the *Z*-axis was observed to be insignificant (in the order of a few microns) due to the optimization of the print process in the *Z*-axis, as discussed in previous sections. Plus, it is noteworthy that while adhering the PDMS device to the glass substrate, gentle pressure was manually applied to the PDMS in order to bond it to the plasma-treated glass slide. This slightly compressed the PDMS, and the minute gap between two Z planes of the MICS and MESS channels could be aligned. However, as mentioned in previous research, the curing condition, including heating time and temperature, can modify the physical properties of PDMS and its performance in bonding to glass or PDMS substrates [15,32].

To demonstrate the functionality of the composite microfluidic MESS/MICS mold, the best set of molds which had the fewest misalignments were used for the preparation of PDMS. The fabrication of the PDMS device followed the procedure explained in the Materials and Methods. The devices were first tested by infusing a food dye to verify that the device had acceptable bonding to the glass slide. Then, two different aqueous solutions, food dye and water, with the same concentrations were infused into the microfluidic channels, and the profile of the mixing was observed, as shown in Figure 12. By this way, we could validate the proposed approach in this paper and demonstrate its efficacy.

## 4. Perspective

Most reported fabrication techniques in mold fabrication employed photolithography, SU8 photoresist on a glass or silicon substrate, in order to transfer micro-features, including channels, pillars, and other features, to the cast material, such as PDMS, to produce microfluidic devices [33,34]. There have been studies that have reported the direct use of a silicon wafer with processed SU8 to generate microfluidic devices that can offer small-sized features; however, the fabrication process is costly and labor intensive [19]. By implementing the proposed technique in this paper, simple and complicated microfluidic molds can be generated, and the integration and assembly of modular microfluidic devices is possible. In addition, with emerging technologies, including micro-/mesoscale sensors [35] and actuators [36], and advanced fabrication methods [37], the integration of a modular mold fabrication technique can help with generating complex microfluidic devices. Although the focus of the proposed fabrication technique was on generating PDMS microfluidic devices, the approach is not limited to the microfluidics application. Wearable sensors [38] and 4D printing materials [39] can take advantage of the proposed technique in connecting microscale and mesoscale features in their wide applications thanks to the advancement of 3D printing technology in providing high-resolution capabilities and various choices of materials, including but not limited to bio-compatible and flexible materials.

Considering the benefits of connecting micro-/mesoscale features in the 3D-printed mold, matching different resolutions of features, such as XY plane and *Z* axis, can be challenging, especially at the interface of micro and mesoscale features. To overcome this challenge, a set of calibration processes that achieve matching resolution properties, geometrical alignment of features in different axes, and utilize high-resolution digital microscopes for accurate measurements are required in order to achieve an optimized connection between microscale and mesoscale features in a single device. One of the factors that can hinder the fabrication of modular devices with micro-/mesoscale features is the accessibility to high-resolution 3D printers and characterization equipment. Recently, many research labs and fabrication service facilities have provided access to conventional 3D printers which do not have the adequate resolution for printing many microscale features. However, by integrating the proposed technique and procedure, two 3D printers with similar 3D printing technology (such as SLA) but different printing resolutions can be used to fabricate modular devices, including microfluidic molds. For future studies, utilizing different types of 3D printing materials and technologies to understand the possibility of generating modular devices and molds, integrating high-resolution features created with photolithography into meso-/microscale prints such as diced silicon devices, implementing nanomaterials into the modular devices to produce sensors and actuators, and integrating 3D-printed nanostructures along with micro-/mesoscale features can be the point of researchers’ interest to investigate and expand the proposed technique.

## 5. Conclusions

In this paper, a new technique for the design and fabrication of microfluidic PDMS molds was introduced in which SLA 3D printing technology was used to print MICS features along with MESS features. This technique requires two different SLA printers, one with a higher resolution to be able to print MICS features and another with a higher throughput to print MESS features. Implementing features with two different scales in a single device has been challenging in various approaches of fabricating molds in microfluidics applications since it often requires labor-intensive and complicated processes. Therefore, using a modular technique helps with addressing this existing challenge. As discussed in this paper, the proposed technique comprises calibration processes for identifying limitations with 3D printers in printing simplified designs of target features based on the CAD file provided, delineating micro-/meso-features from the CAD file and proceeding with sectioning them. Then, the sectioned MESS part is printed, post-processed, and the critical dimensions of the sectioned spaces that interface with the MICS part are measured. Once the measurements are collected, they are applied to the MICS part by modifying the critical dimensions for improved mating. The modified microscale part is printed with the higher-resolution printer, post-processed, and mated with the MESS part. The process of ensuring the proper alignment of the MICS part with the MESS part was extensively discussed. Finally, the composite mold was used to successfully mold the serpentine mixer device from PDMS to demonstrate the concept proposed in this paper. There are some limitations with this approach, including the access to both high- and lower-resolution 3D printers, but this can be a motivation to try other fabricating technologies to build a modular mold. Although only MICS and MESS features were demonstrated here, this technique could likely be extended by implementing nanostructures in the MICS/MESS features to connect the three different scales (nano, micro, and meso) in a single device.

## Figures and Tables

**Figure 1 micromachines-14-01363-f001:**
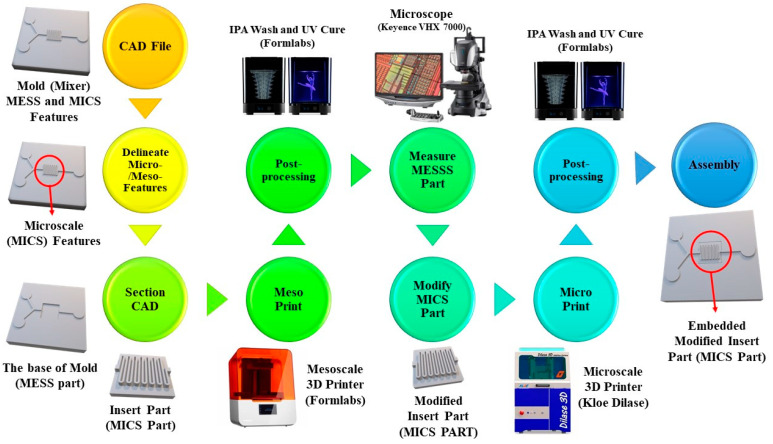
Process flow for fabricating microfluidic molds using both MICS/MESS SLA 3D printers. In summary, after delineating MESS/MICS features from the CAD file, the MICS part would be sectioned from the MESS part. Then, the MESS 3D printer would be utilized for printing the associated part. After post-processing and measuring the MESS part, geometrical modifications would be applied to the MICS part design, and print and post-processing follow accordingly. Finally, the printed parts, MESS and modified MICS, would be assembled to form the microfluidic mold.

**Figure 2 micromachines-14-01363-f002:**
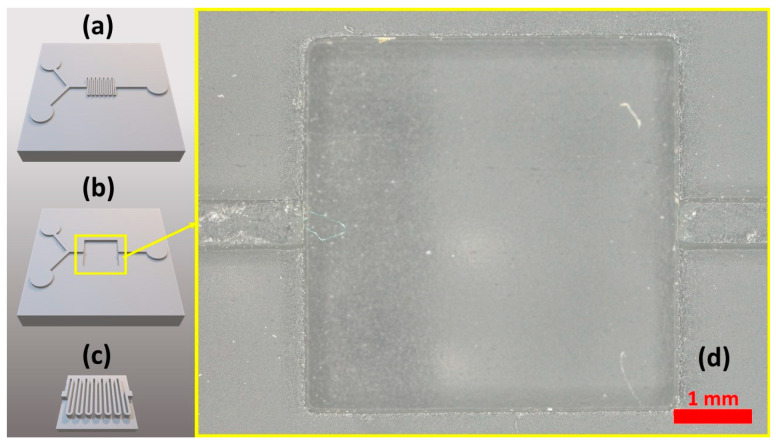
(**a**) CAD image of mixer mold. (**b**) Base and (**c**) insert parts after delineating and sectioning MICS/MESS features. (**d**) Magnified image of the base part.

**Figure 3 micromachines-14-01363-f003:**
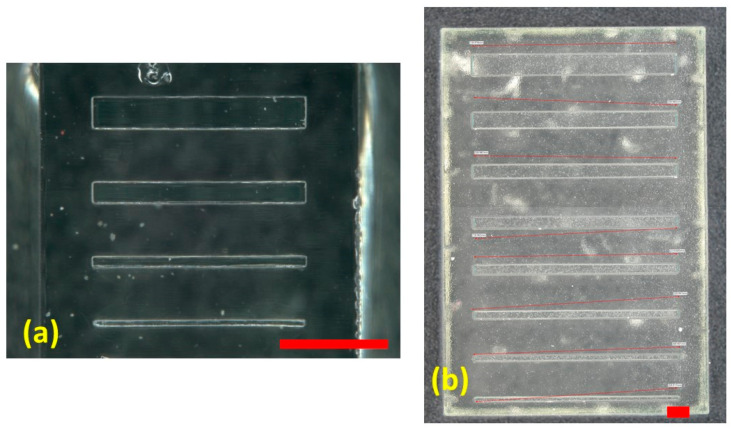
Straight channels printed with a (**a**) MICS printer, channels with widths of 300, 200, 100, and 50 µm (top to bottom), and a (**b**) MESS printer, channels with width of 1000, 750, 600, 500, 400, 300, 200, and 100 μm (top to bottom). All red scale bars are 1 mm.

**Figure 4 micromachines-14-01363-f004:**
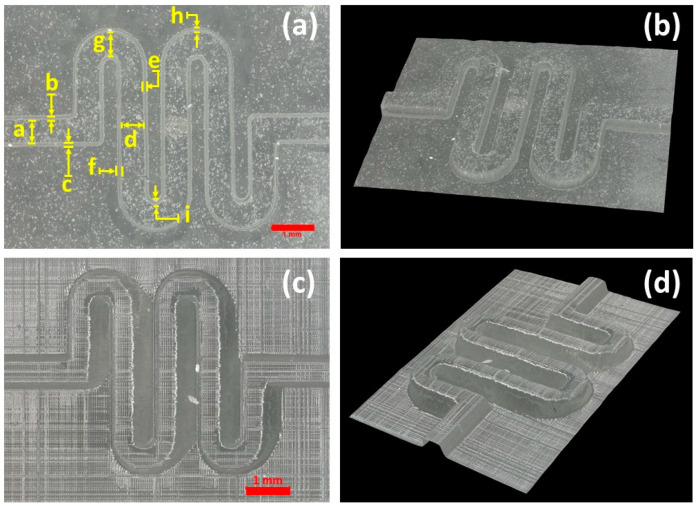
Mixer molds printed using the MESS printer. (**a**) 0-degree orientation of the mixer on the build platform and critical dimensions measured. Dimension values are demonstrated in Table 2. (**b**) Surface profilometry of 0-degree mixer mold. (**c**) 90-degree mixer mold. (**d**) 90-degree mixer mold surface profilometry.

**Figure 5 micromachines-14-01363-f005:**
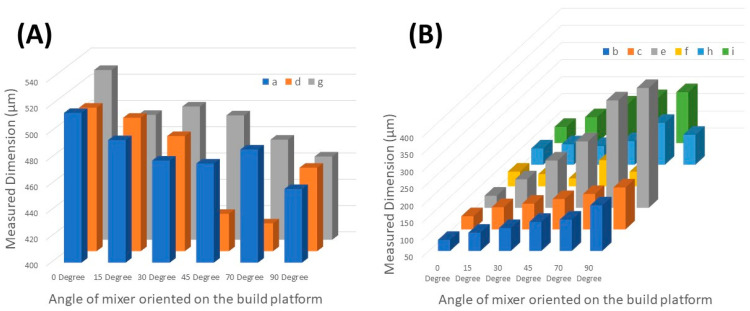
Channel width and side profile widths of serpentine channels printed using the MESS printer. (**A**) Channel widths in x-direction (a), y-direction (d), and curved channels (g). (**B**) Channel side profile widths for x-direction top (b) and bottom (c), y-direction right side (e) and left side (f), and curved channel outer (h) and inner (i) side of radius.

**Figure 6 micromachines-14-01363-f006:**
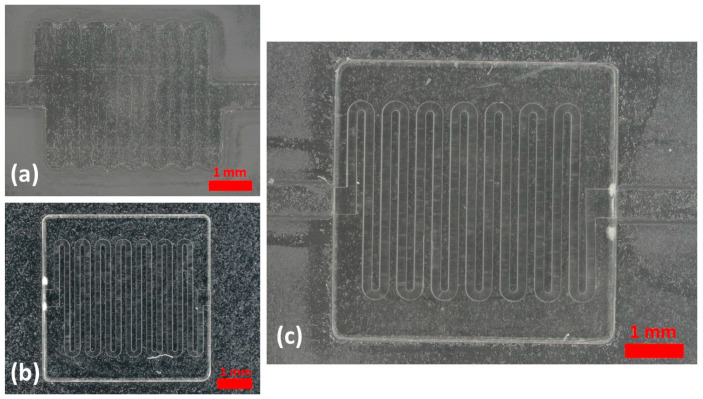
Pictures of the insert part. (**a**) The insert part and serpentine channels with a 150 μm width printed with a MESS 3D printer. As can be seen, the channels are not defined due to the limited resolution of the MESS printer. (**b**) The insert printed with a MICS 3D printer with applied dimension modifications after microscopic measurement of the base. (**c**) Mated MESS/MICS parts (the base and insert).

**Figure 7 micromachines-14-01363-f007:**
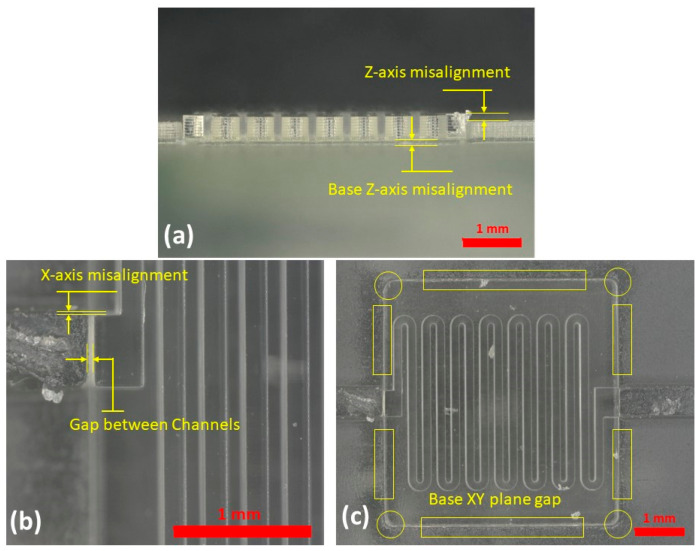
Assembled MICS and MESS parts demonstrating the misalignment. (**a**) *Z*-axis misalignment of MESS with MICS channels and the insert and base misalignment in *z*-axis. (**b**) *X*-axis misalignment and the gap between MICS and MESS channels. (**c**) Gaps measured between the insert and base parts in XY plane.

**Figure 8 micromachines-14-01363-f008:**
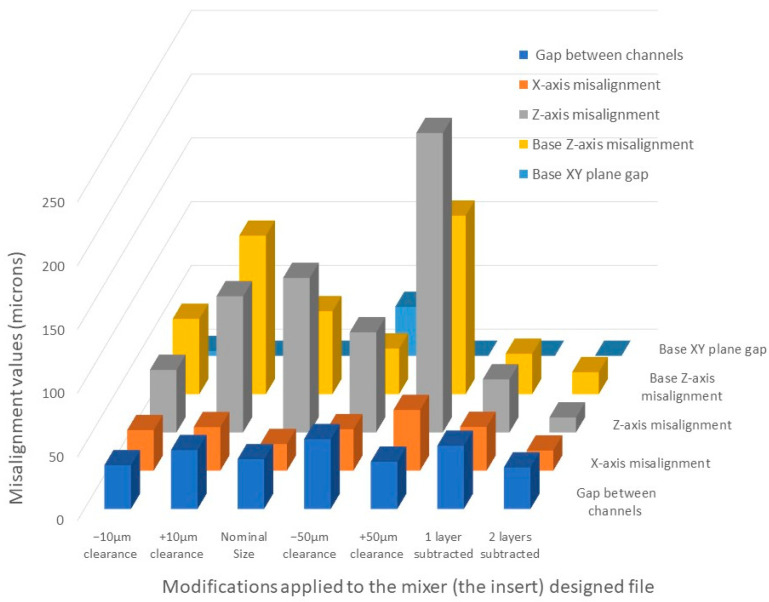
Misalignment chart based on data presented in Table 4.

**Figure 9 micromachines-14-01363-f009:**
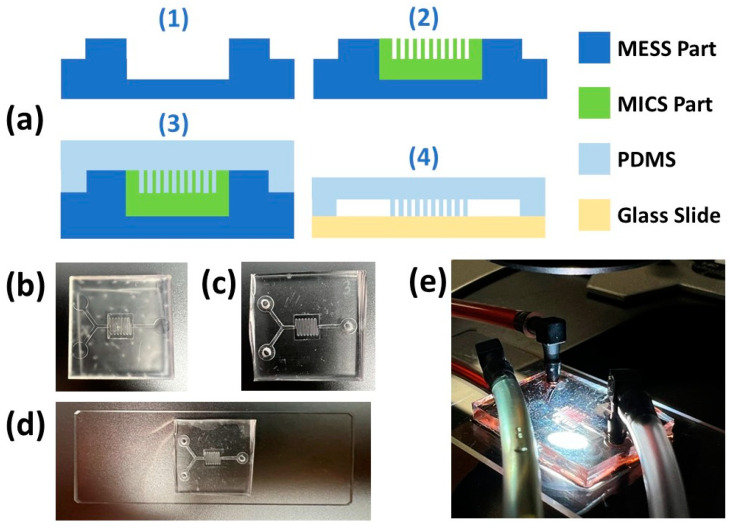
(**a**) Schematic representation of MICS/MESS mold fabrication and assembly (**1**,**2**) and PDMS device fabrication, including casting PDMS into the MICS/MESS mold and processing it (**3**) and bonding the cured and peeled PDMS to the plasma-treated glass slide (**4**). (**b**) Assembled 3D-printed MICS/MESS mixer mold. (**c**) Peeled-off PDMS mixer with punctured inlets and outlet. (**d**) Bonded PDMS mixer device on the plasma-treated glass slide. (**e**) Serpentine mixer PDMS device with connected tubes in order to deliver food dye aqueous solutions.

**Figure 10 micromachines-14-01363-f010:**
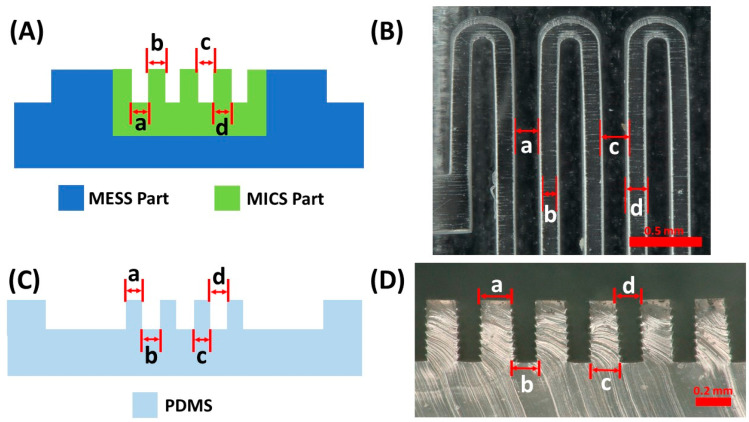
Presentation of characterized features for the 3D-printed MICS part and cast PDMS device. (**A**,**B**) Illustration of measured features in the MICS part: (a) gap between channels—bottom surface, (b) channel width—top surface, (c) gap between channels—top surface, (d) channel width—bottom surface. (**B**) Measured features on the printed MICS part. An example of conducted measurements is shown in the Appendix A. (**C**,**D**) Illustration of measured features in PDMS device: (a) PDMS wall width—top surface, (b) channel width—bottom surface, (c) PDMS wall width—bottom surface, (d) channel width—bottom surface. (**D**) Cross-section view of measured features on the sliced PDMS device.

**Figure 11 micromachines-14-01363-f011:**
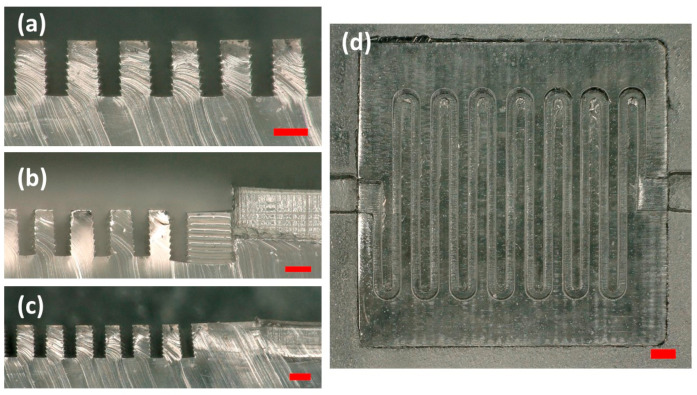
PDMS serpentine mixer after peeling off from the MICS/MESS 3D-printed mixer mold. (**a**) Cross-sectioned view of sliced PDMS channels. (**b**,**c**) Cross-sectioned view of two different PDMS devices molded from an assembled mold with *Z*-axis misalignment at the interface of MICS and MESS channels (**b**) and a device with less *Z*-axis misalignment (**c**). (**d**) Top view of serpentine mixer channels of PDMS device. The scale bar is 200 µm.

**Figure 12 micromachines-14-01363-f012:**
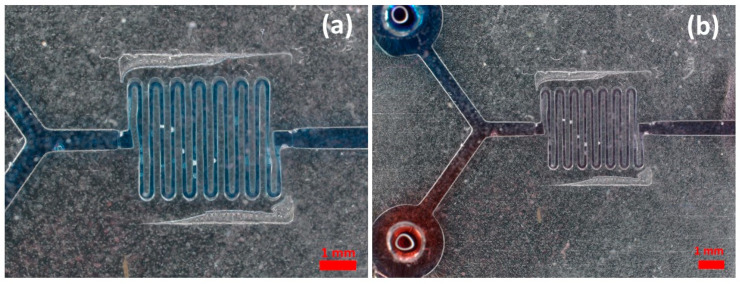
Microfluidic PDMS serpentine mixers fabricated out of composite mixer molds. (**a**) Single food dye infused in the device with a 5 µL/min flow rate. (**b**) PDMS mixer mold demonstrating mixing functionalities. Red aqueous and blue aqueous solutions both had the same flow rates, 5 µL/min.

**Table 1 micromachines-14-01363-t001:** Designed dimensions vs. measured dimensions for width of straight channels printed with MESS/MICS printers. Mean ± standard deviation. Three samples were printed for each dimension, and five measurements were taken for each sample. Dimensions in bold font are printed in both printers. NP: not printed, MICS: microscale, MESS: mesoscale.

Designed Width Dimension (μm)	Average Printed Width Dimension (μm)	Error from Design (%)	Printer Type
50	47.66 ± 3.00	4.66	MICS
**100**	96.8 ± 3.21	3.2	MICS
**200**	195.73 ± 7.43	2.13	MICS
**300**	292.86 ± 2.72	2.37	MICS
**100**	128.93 ± 15.56	28.93	MESS
**200**	197.53 ± 2.66	1.23	MESS
**300**	292.8 ± 1.31	2.4	MESS
400	405.66 ± 0.92	1.41	MESS
500	507.33 ± 7.76	1.46	MESS
600	612.8 ± 4.88	2.13	MESS
750	762 ± 9.27	1.6	MESS
1000	1003.46 ± 6.33	0.34	MESS

**Table 3 micromachines-14-01363-t003:** Print time of the mixer mold, including entire mold, base, and insert, using two different 3D printers. The number of layers changes for the insert part since the alignment of the insert and the base after mating was explored and modified. The data and discussion for the alignment are presented in Section 4.

Features	Number of Layers	Printer	Print Time
Insert part	18	Dilase 3D	1 h 58 min
Insert part with a subtracted layer from its base	17	Dilase 3D	1 h 1 min
Insert part with a subtracted layer from its base and a layer from serpentine channel	16	Dilase 3D	57 min 57 s
Entire Mixer Mold	69	Dilase 3D	38 h 52 min
Entire Mixer Mold	69	Formlabs	24 min
Base of Mixer Mold	69	Formlabs	23 min

**Table 4 micromachines-14-01363-t004:** Misalignment values for MESS and MICS parts with different clearances applied to the MICS parts. Mean ± standard deviation. Three samples were printed for each clearance, and five measurements were taken for each sample.

Features	−10 μm Clearance	+10 μm Clearance	Nominal Size	−50 μm Clearance	+50 μm Clearance	1 Layer Subtracted	2 Layers Subtracted
Gap between channels	34.34 ± 20.12	46.18 ± 14.12	39.03 ± 7.89	54.78 ± 27.79	37.05 ± 22.92	49.6 ± 13.85	32.59 ± 11.14
*X*-axis misalignment	31.82 ± 18.95	34.21 ± 32.05	20.92 ± 11.27	32.47 ± 27.15	47.62 ± 25.97	34.34 ± 25.18	15.84 ± 12.68
*Z*-axis misalignment	49.06 ± 15.61	106.83 ± 43.02	121.16 ± 25.28	78.5 ± 17.37	234.83 ± 73.68	41.66 ± 102.06	−11.76 ± 14.94
Base *Z*-axis misalignment	59.00 ± 4	124.25 ± 34.67	65 ± 4.32	35.75 ± 13.79	140 ± 37.47	31.66 ± 19.36	17.19 ± 17.61
Base XY-plane gap	3.83 ± 12.71	0	0	38.46 ± 16.68	0	0	0

**Table 5 micromachines-14-01363-t005:** Measured dimensions of the MICS part and PDMS device. Designed dimensions for all the features mentioned in this table were 150 µm. To better understand the measurements, refer to Figure 10. Three MICS parts and the PDMS molded from them were utilized as measurement samples, and five measurements were taken for each feature.

Measured Features for MICS Part	Gap between Channels Bottom Surface (a)	Channel Width Top Surface (b)	Gap between Channels Top Surface (c)	Channel Width Bottom Surface (d)
Mean ± Standard Deviation (µm)	150.2 ± 6.40	99.66 ± 2.19	199.86 ± 2.92	152.26 ± 5.10
**Measured Features for PDMS Device**	**PDMS Wall Width** **Top Surface (a)**	**Channel Width Bottom Surface (b)**	**PDMS Wall Width Bottom Surface (c)**	**Channel Width** **Top Surface (d)**
Mean ± Standard Deviation (µm)	155 ± 3.14	117.66 ± 7.32	177.26 ± 3.10	127.33 ± 8.51

## Data Availability

Data is contained within the article or Appendix A.

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
