# Peer review of "A Non-Sacrificial 3D Printing Process for Fabricating Integrated Micro/Mesoscale Molds"

_micromachines, 2023, doi:10.3390/mi14071363_

Round 1

Reviewer 1 Report

The article proposed a non-sacrificial 3D printing process for microscale molds. The article can be considered for publication after major revising the following questions. The comments are below.

1) Through the figures in the manuscript, it can be found that there is obvious dust on the PDMS. Why did the authors not clean it? Microfluidic chips are widely used in chemical and biological reactions, and if dust cannot be removed, the application of this processing method will be greatly limited.

2) PDMS is an elastic material. The PDMS may shrink or expand when removed from the mold. The authors should compare the PDMS with the mold size to get a general rule. It can further optimize dimensional accuracy.

3) In Section 1 Introduction, author analyzed two methods of 3D printing including FDM and SLA. There exists capital 3D printing called DLP from many articles. I recommend introducing DLP for microfluidic chips and comparing it with the mentioned methods. Biosensors and Bioelectronics, 2023, 230, 115283.

4) In Figure 1, the author utilized two 3D printers for fabricating microfluidic molds. Is there any difference in process between two printers? If yes, can modifications on the molds made from the first printer be applied to the second printer?

5) In Figure 2, CAD image of mixer mold show that the final mold consists of two parts including base and insert. The sealing performance is a concern. Do the changes in fluid flow rates and residence times cause leakage?

6) Compared with the overall 3D printing mold, what are the advantages of this embedded mold? I suggest comparing the embedded mold with the mold in these articles. Micromachines (Micromachines), 2021, 12(11), 1380

7) In Table 1, the error of MESS increase obviously when design size changes from 200μm to 100 μm. What are the main reasons for this? When using the MESS method for mold printing, will the error continue to increase when the size of the channel is less than 100μm?

8) In Figure 3,4, the photographs of molds are captured by camera. The statistical method of size is not comprehensive. It can be seen that the straightness of the channel is not good. The authors should measure straightness or select more measuring points. Moreover, the authors do not give a cross-section characterization of the channel, which is also the focus of the evaluation of manufacturing performance. Authors can refer to Analytical Methods, 2018, 10, 2470. Military Medical Research, 2022, 9, 8.

9) In Figure 4, the photographs of mixer molds are exhibited. Please introduce the material of molds including the model of resin, heat tolerance, mechanical properties, etc.

10) In Figure 5, the angle orientation of molds was studied. SLA 3D printing is printed layer by layer. The Z-axis accuracy is highly dependent on the printer's accuracy. Therefore, the surface of the mold printed at different inclination angles inevitably has stepped lines. Polishing and other post-processing may improve the surface finish, but it will cause a waste of time. Does it make sense to discuss a lot about inclination angles?

11) In Figure 6, the pictures of insert part were exhibited. I recommend adding local characterization from metallographic microscopy.

12) In Figure 9b, it can be seen that the red liquid has leaked in the embedded part. What are the main causes of leaks?

13) The authors did not provide parameters such as the flow rate when mixing the liquid, which is essential for mixing experiments. In addition, the authors did not evaluate the mixing efficiency of micromixers, which affects the application value of micromixers. Authors can refer to Analytica Chimica Acta, 2021, 1155, 338355. Journal of Materials Chemistry B, 2023, 11, 1978-1986. ACS Omega, 2021, 6(45), 30779-30789. 

Author Response

The authors would like to thank you for your kind review and comments on the manuscript.

Please see the attachment for the point-by-point response to your comments. 

Your comments are in blue font and our response is in black font. 

Reviewer 2 Report

The paper introduces an innovative non-sacrificial methodology for the fabrication of microscale and mesoscale features in microfluidics, employing Stereolithography (SLA) 3D printers. The authors ambitiously seek to surmount the inherent limitations and complexities associated with traditional soft lithography techniques by harnessing the design freedom, rapidity, and cost-effectiveness offered by 3D printing technology. The focal point of their proposition resides in a meticulously designed modular microfluidic mold assembly process, enabling the efficient production of intricate and time-intensive micro/mesoscale devices. Comprehensive discussions within the paper encompass the intricacies of the process flow, optimization of print time, refinement of feature resolution, precise alignment of modular devices, and a judicious analysis of the associated advantages and limitations inherent to the proposed technique.

However, prior to the acceptance of this paper for publication, there are several salient points that necessitate clarification. For instance, an incongruity arises in the abstract, wherein the prominence accorded to PDMS seems to eclipse the main focus on 3D printing, as explicitly emphasized in the introduction. The authors are thus urged to rectify this inconsistency to enhance the cohesiveness of their work. Additionally, a meticulous review of the manuscript reveals some inaccuracies pertaining to the formatting of the references (comas are needed). It is imperative that these inconsistencies are promptly addressed to ensure the readability and comprehensibility of the manuscript.

With regards to Figure 1, it is indispensable that labels be added to provide a clear and concise description of both MICS and MESS, as their precise identification within the figure is currently ambiguous and perplexing to readers. Moreover, from a cursory examination of Figure 1, it remains unclear what modifications have been made to the "Modified Insert Part," as it appears indistinguishable from the "Insert Part." The caption accompanying Figure 1 should be suitably revised to expound upon this distinction and provide a comprehensive depiction of the entire assembly, allowing readers to grasp the final configuration of the component at one sight.

Given the nature of the paper, which focuses on molding technology, it is crucial to deliberate upon the unmolding procedure. Elucidating how the molded part was successfully extracted from the mold, along with any attendant challenges encountered and potential recommendations or insights gained, would provide valuable insights for the scientific community. Inclusion of such information is highly recommended to enhance the thoroughness and practical applicability of the paper.

Furthermore, it is strongly advised to incorporate a Perspective section that offers recommendations on adapting this technique for the fabrication of high-resolution molds employing photolithography and sectioned wafers via the utilization of a dicing saw. This section should include a comprehensive exploration of the potential future applications and benefits that can be derived from the ability to manufacture modular microfluidic molds using 3D printing technology, specifically in conjunction with high-resolution capabilities. Additionally, it is important to discuss the challenges and limitations that may arise with the current approach and suggest avenues for improvement or optimization, such as material selection, resolution enhancement, surface treatment, and compatibility with diverse fluids and experimental conditions. Finally, an examination of the potential integration of the proposed technique with other emerging technologies, including microscale sensors, actuators, and advanced fabrication methods, should be explored to further augment the functionality and versatility of microfluidic devices.

By addressing these constructive comments and augmenting the manuscript accordingly, the paper will undoubtedly be positioned for publication, enriched with a more comprehensive perspective on the research and its future implications.

Sufficiently good.

Author Response

(The authors gave the same response as above.)

Round 2

Reviewer 1 Report

The article proposed a non-sacrificial 3D printing process for microscale molds.

The article can be considered for publication in "Micromachine" after revising the following questions. The comments are below.

1) In the first review, the author suggested that the main reason for leakage is due to imperfect curing of PDMS. I recommend replicating the experiment in Figure 12B to improve the reliability of PDMS curing.

2) In the first review, the author explained the angle orientation of molds. However, it seems that the author did not respond to the step pattern of molds caused by angle. “In Figure 5, the angle orientation of molds was studied. SLA 3D printing is printed layer by layer. The Z-axis accuracy is highly dependent on the printer's accuracy. Therefore, the surface of the mold printed at different inclination angles inevitably has step patterns. Polishing and other post-processing may improve the surface finish, but it will cause a waste of time. Does it make sense to discuss a lot about inclination angles?” Please discuss this in the second review.

Author Response

The authors thank you for your comments on the revised manuscript.

Please see the attachment for our response to your comments and suggestions.

Your comments and suggestions are in blue font and our response is in black font. 
